# Ultrasound Evaluation of Internal Jugular Venous Insufficiency and Its Association with Cognitive Decline

**DOI:** 10.3390/diagnostics15111427

**Published:** 2025-06-04

**Authors:** Jiu-Haw Yin, Nai-Fang Chi, Wen-Yung Sheng, Pei-Ning Wang, Yueh-Feng Sung, Giia-Sheun Peng, Han-Hwa Hu

**Affiliations:** 1Division of Neurology, Department of Internal Medicine, Taipei Veterans General Hospital, Hsinchu Branch, Hsinchu City 310403, Taiwan; 2Department of Neurology, Tri-Service General Hospital, National Defense Medical Center, Taipei 114202, Taiwan; sungyf@mail.ndmctsgh.edu.tw (Y.-F.S.); tsghpeng@gmail.com (G.-S.P.); 3Neurological Institute, Taipei Veterans General Hospital, Taipei 112201, Taiwan; naifangchi@gmail.com (N.-F.C.); wysheng436@gmail.com (W.-Y.S.); linda2860@gmail.com (P.-N.W.); 4Department of Neurology, School of Medicine, National Yang Ming Chiao Tung University, Taipei 112304, Taiwan; 5Division of Neurology, Department of Internal Medicine, Taipei Veterans General Hospital, Taoyuan Branch, Taoyuan City 330023, Taiwan; 6Department of Neurology, Taipei Medical University-Shaung Ho Hospital, New Taipei City 235041, Taiwan; hanhwa@hotmail.com

**Keywords:** internal jugular vein, internal jugular venous insufficiency, cognitive impairment, neuropsychological examination, ultrasonography

## Abstract

**Background:** Prior studies have shown an association between jugular venous reflux and age-related neurological conditions, including cognitive decline and potentially incident dementia. However, a relationship between internal jugular vein (IJV) outflow disturbance and cognitive impairment has yet to be elucidated. This study evaluates the relationship between impaired IJV drainage and cognitive function. **Methods**: We recruited a prospective sample of 106 participants with subjective memory complaints. Subjects underwent neuropsychological assessments and ultrasound examination of IJV, including time-averaged mean velocity (TAMV) and the cross-sectional area of the IJV at the middle (J2) and distal (J3) segments. Impaired IJV drainage was defined by either of the following: (1) TAMV < 4 cm/s at the J2 or J3 segment on either side, or (2) IJV lumen collapse during inspiration at the J2 segment on either side. **Results**: The impaired cognition group had a significantly higher prevalence of both impaired flow velocity and impaired IJV drainage compared to the normal cognition group (34% vs. 16%, *p* = 0.032; 68% vs. 30%, *p* < 0.001). Furthermore, the impaired IJV drainage group demonstrated lower scores across all neuropsychological tests, with statistical significance observed in the Mini-Mental State Examination (median (IQR) 27 vs. 29, *p* = 0.013), Montreal Cognitive Assessment (median (IQR) 23 vs. 26, *p* < 0.001) and Chinese Version of the Verbal Learning Test (median (IQR) 23.5 vs. 27, *p* = 0.024). Notably, incorporating IJV lumen collapse during deep inspiration into the definition of impaired IJV drainage further increased its prevalence in the impaired cognition group. **Conclusions**: Our results revealed that the impaired cognition group exhibited a higher prevalence of impaired outflow in the bilateral IJV, while the impaired IJV drainage group scored lower on all neuropsychological tests compared to the normal group. These findings support the hypothesis that impaired IJV drainage is correlated with global cognitive decline.

## 1. Introduction

The internal jugular vein (IJV) is the main pathway for cerebral venous drainage and any outflow impairment may lead to cerebral venous hypertension or congestion [1,2]. A growing body of research suggests that alterations in cerebral venous circulation may be associated with cognitive decline and various neurodegenerative diseases, including Alzheimer’s disease (AD) and amyotrophic lateral sclerosis [3].

Impaired IJV drainage refers to the restriction or obstruction of blood flow through the IJV, with jugular venous reflux (JVR) being the condition most commonly studied in clinical settings. JVR is characterized by reflux in the IJV during Valsalva maneuvers (VMs) or during spontaneous rest, and is more common in the elderly. Sustained JVR may cause a retrograde pressure transmission into the central nervous system, thereby hampering brain parenchyma and circulation [3]. JVR is potentially a clinically important hemodynamic abnormality, which has been shown to influence cerebral blood flow and increase cerebral venous pressure by stagnation or reversal of the flow in the IJV [4]. JVR has been found to be associated with age-related neurological conditions such as normal pressure hydrocephalus [5], transient global amnesia [6], white matter lesions [7] and leukoaraiosis [8]. Mild cognitive impairment (MCI) is regarded as a transitional period between the normal cognitive decline of healthy aging and dementia, characterized by objective impairment in cognition that is not severe enough to require help with daily activities [9]. A recent study identified an association between JVR and global cognitive function, and a potential link with incident dementia [10]. However, the extent to which jugular venous outflow disturbance contributes to cognitive impairment has yet to be elucidated.

Duplex ultrasonography of the IJV is a reliable method for central venous pressure monitoring, providing a portable, cost-effective, and non-invasive assessment of the neck veins [11]. Based on this approach, a recent study found a correlation between spontaneous echo contrast or IJV velocity and the development of AD [12]. However, there is currently no consensus on extracranial venous blood flow volume measurement and a standardized reference range for diagnosing impaired IJV drainage [13]. The present study examines whether impaired IJV drainage is related to cognitive function by evaluating the IJV outflow and collapsibility of the IJV lumen in response to deep inspiration in patients with cognitive impairment.

## 2. Materials and Methods

### 2.1. Subjects

Participants were selected from Taiwanese residents who were first-time admissions to the memory clinic at Taipei Veterans General Hospital from May 2022 to May 2023 due to subjective memory complaints. Neurologists performed clinical and neurologic evaluations on all participants. Subjects eligible for participation were aged 55 years or older, had a Clinical Dementia Rating score of ≤2 (to confirm ability to safely and effectively perform the IJV collapse test), and were willing to receive neck duplex ultrasonography and transcranial Doppler study. Exclusion criteria included other etiologies such as depression or other psychiatric disorders, vitamin B12 or folic acid deficiency, neurosyphilis, long-COVID, lack of cooperation with quantitative data analysis, along with vascular dementia as determined by routine CT/MRI interpretation and clinical assessment to avoid experimental bias (Appendix A). Study approval was obtained from the Taipei Veterans General Hospital’s institutional review board, and this cross-sectional study was conducted in accordance with institutional guidelines. Written informed consent was acquired from all participants.

### 2.2. Clinical Assessments

Cardiovascular risk factors were either measured or assessed through self-reporting. Body mass index (BMI) was calculated as weight in kilograms divided by height in meters squared. Hypertension was defined as systolic blood pressure ≥ 140 mmHg, diastolic blood pressure ≥ 90 mmHg, or treated or self-reported hypertension [14]. Type 2 Diabetes mellitus (DM) was defined by either self-reporting of current DM medication or measurement of hemoglobin A1c ≥ 6.5% [15]. Dyslipidemia was defined as a total cholesterol level ≥ 200 mg/dL, triglyceride level ≥ 150 mg/dL, low-density lipoprotein cholesterol level ≥ 130 mg/dL, high-density lipoprotein cholesterol level < 40 mg/dL, or self-reported hyperlipidemia. Chronic kidney disease (CKD) was defined according to the estimated glomerular filtration rate (eGFR) ≤ 60 mL/min/1.73 m^2^ [16]. Thyroid dysfunction was defined as the presence of either overt or subclinical hyperthyroidism or hypothyroidism [17]. The presence of stroke or myocardial infarction history was determined by self-report or medical record.

### 2.3. Color-Coded Duplex Ultrasonography

Neck color-coded duplex sonography was performed in all enrolled participants using a 7 MHz linear transducer (IU 22; Philips, New York, NY, USA). It was performed consistently by a single technician with more than 15 years of experience who was blinded to the subjects’ characteristics. During examination, subjects were asked to lie in a head-straight, flat supine position after a quiet 10 min rest. An ultrasound of the extracranial venous system was performed according to the previously reported process [4,6,18]. In brief, the IJV’s time-averaged mean velocity (TAMV, cm/s) and the lumen cross-sectional area (CSA, cm^2^) were recorded at its middle (J2) and distal (J3) segments (Figure 1). The junction between J2 and J3 is located where the common facial vein drains into the IJV. [18,19]. The segment of the IJV above this junction is referred to as J3, while the segment below it is referred to as J2 [6]. We acquired the TAMVs by directing the Doppler cursor parallel to the vein with the gate adjusted to comprise the entire lumen. We measured the TAMVs with the IU 22 built-in software (6.3.7.745) and included at least three cardiac cycles on the Doppler spectrum. The probe was then rotated 90◦ according to the same IJV segments to measure the CSAs. The CSA was measured three times using B-mode imaging and then averaged for later analysis. The TAMV of IJV was recorded in a resting state with free breathing. Routine cervical arterial examination, including carotid duplex sonography of common carotid arteries (CCA), internal carotid arteries (ICA) and vertebral arteries (VA), as well as transcranial Doppler study of middle cerebral arteries (MCA) were also performed on all enrolled patients.

### 2.4. IJV Drainage Determination

The change in IJV lumen CSA was recorded during brief apnea following two respiratory statuses: (1) normal expiration (baseline), and (2) deep inspiration. All subjects were asked not to strain during breath-holding to avoid increasing intrathoracic pressure. The method was similar to a previously described method, with slight modification [20]. We defined IJV TAMV of <4 cm/s as abnormal based on our previous study of transient global amnesia patients and another study of Alzheimer’s disease patients [6,12]. A CSA of IJV shrinking to less than 1/3 of its original size during inspiration was defined as IJV collapse (Figure 2). Impaired flow velocity was defined as TAMV < 4 cm/s at the J2 or J3 segment on either side. Furthermore, impaired IJV drainage was defined by one of the following features: 1. the above-mentioned outflow impairment, or 2. IJV lumen collapse during inspiration at the J2 segment on either side.

### 2.5. Cognitive Function Assessment

All study participants were subjected to an in-person neuropsychological examination administered by trained interviewers. Global cognitive performance was assessed using the Mini-Mental State Examination (MMSE) [21] and Montreal Cognitive Assessment (MoCA) [22]. Additional verbal memory domain was assessed using the Chinese Version of the Verbal Learning Test (CVVLT) [23]. Clinically significant cognitive impairment was defined by an MMSE score of ≤24.

### 2.6. Statistical Analysis

Statistical analysis was undertaken using SPSS 24 (IBM, Armonk, New York, NY, USA). Continuous variables were expressed as mean ± SD or median with interquartile range (IQR) as applicable. Categorical variables were expressed as percentages. In continuous data, we used the nonparametric statistics Wilcoxon rank sum test for differences between groups, and the chi-square test for categorical variables. The odds ratios (ORs) and their 95% confidence interval (CI) were used to present the effect of a single variable on cognitive tests. In a multivariable logistic regression, we used a backward stepwise selection model to identify the main factors affecting the cognitive function. A *p*-value less than 0.05 was considered statistically significant.

## 3. Results

### 3.1. Baseline Clinical Characteristics of Impaired Cognition and Control Group

A total of 106 subjects (67 female) were enrolled. The mean age of the group was 71.5 ± 8.0 years; the mean education age was 13.1 ± 3.8 years. The group included 50 cases (47.2%) with cognitive impairment and 56 normal cases. MoCA and MMSE test scores were all consistent with the disease distribution. Compared to the normal group, the impaired cognition group were older and had fewer years of education (11.3 ± 4.5 vs. 14.4 ± 2.9, *p* < 0.001) and a significantly lower ratio of dyslipidemia (Table 1). Furthermore, the Spearman correlation coefficient between the basic information of the research sample and its neuropsychological test scores showed a significant negative correlation between age and neuropsychological test scores (MMSE: r = −0.564; MoCA: r = −0.701, *p* < 0.0001) but a significant positive correlation between years of education and neuropsychological test scores (MMSE: r = 0.577; MoCA: r = 0.689, *p* < 0.0001). In other words, advanced age was associated with lower neuropsychological test scores, while more extensive education corresponded to higher neuropsychological test scores (Appendix A).

### 3.2. IJV Flow Velocity and Drainage Between Impaired Cognition and Control Group

The impaired cognition group had a significantly higher prevalence of impaired flow velocity and IJV drainage compared to the normal group (34% vs. 16%, *p* = 0.032; 68% vs. 30%, *p* < 0.001). Further analysis showed that the impaired cognition group exhibited higher prevalence of TAMV< 4 cm/s at the J3 segments of the bilateral IJVs (left: 34% vs. 18%, *p* = 0.057; right: 12% vs. 2%, *p* = 0.032), particularly on the right side (Table 2). Although the impaired cognition group had a higher proportion of IJV lumen collapse, it did not reach statistical significance. In both the normal and cognitive impairment groups, the intracranial blood flow of the bilateral MCAs was not significantly different, except for a higher pulsatility index (PI) of left MCA found in the impaired cognition group (Appendix A).

### 3.3. Comparison of Neuropsychological Tests Between Impaired IJV Drainage and Control Group

The impaired IJV drainage group had lower scores across all neuropsychological tests compared with the normal group, reaching statistical significance in MMSE (median 27 (IQR, 26~29) vs. median 29 (IQR, 27~30), *p* = 0.013), MoCA (median 23 (IQR, 21~26) vs. median 26 (IQR, 23~28), *p* < 0.001) and CVVLT (median 23.5 (IQR, 21~27) vs. median 27 (IQR, 22~30), *p* = 0.024) (Figure 3). We performed a backward stepwise selection model to determine which confounding variables were associated with cognitive impairment and found that age, years of education, MCA PI, and dyslipidemia were also statistically significant. However, only age (OR:1.07, 95% CI: 1.00~1.13; *p* = 0.0394), education years (OR:0.76, 95% CI: 0.67~0.88; *p* = 0.0001) and impaired IJV drainage (OR:3.84, 95% CI: 1.33~11.09; *p* = 0.0130) were significantly associated with cognitive impairment after multivariate analyses (Table 3). Increased age is positively associated with the risk of cognitive impairment, whereas a higher number of years of education appears to have a protective effect. Nevertheless, impaired IJV drainage was strongly and significantly associated with cognitive impairment.

## 4. Discussion

First, our results are consistent with previous findings of the prevalence of cognition impairment in older people and those with fewer years of education [24,25]. Secondly, the impaired cognition group was found to exhibit a higher prevalence of impaired flow velocity in bilateral IJV. The collapse of the IJV lumen during deep inspiration showed a prevalence in the impaired cognition group, but this was not statistically significant and the prevalence of impaired IJV drainage would be significantly higher if this parameter were combined with the above-mentioned impaired flow velocity as part of the definition of impaired IJV drainage. Third, the impaired IJV drainage group had lower scores across all neuropsychological tests used in this study compared with the normal group. In addition, multivariate analyses showed that age, years of education, and impaired IJV drainage were significantly associated with cognitive impairment.

A recent study identified a correlation between IJV velocity and the development of AD, reporting a TAMV of IQR of 3.9–7.9 cm/s in AD patients [12]. Our previous study on transverse sinus (TS) hypoplasia calculated a TAMV value of 4.12 cm/s by averaging the lower limits of the IJV velocity in both anatomical TS hypoplasia and flow-related TS hypoplasia groups [18]. Accordingly, we adopted a more stringent TAMV standard and used statistical methods to determine cutoff points for TAMV (<4 cm/s). The TAMV and CSA measurements of the IJV not only depend on the respiratory cycle but also retrograde pressure changes caused by the heart. Due to the naturally elliptical form of the internal jugular vein, use of diameter-based CSA measurement of blood flow volume (BFV) produces inaccurate results [13]. In addition, IJV encompasses many tributaries and venous drainage can proceed through alternative routes [26], resulting in significant differences in the flow measurement. In this study, we attempted to compare the differences between the two groups by multiplying TAMV by CSA to calculate BFV, but the results were not statistically significant. In contrast, we found that the impaired cognition group had a higher prevalence of lower outflow in both IJVs compared to the normal group, with statistical significance on the right side. Furthermore, when we incorporated the collapse of the IJV lumen during deep inspiration into the definition of impaired IJV drainage, its prevalence was significantly higher in the impaired cognition group. However, there was no statistical significance in IJV lumen collapse between the two groups, although the impaired cognition group had a higher proportion of IJV lumen collapse.

The CSA of the IJV is variable, as it is a compliant structure influenced by factors such as age, gender, laterality, and cervical location [27]. Measurements can also fluctuate with patient positioning, respiration, and cardiac function [28]. The venous wall is significantly more distensible than the arterial wall, and this has important physiological implications. The distensibility of the IJV is a major determinant of cerebral venous drainage and helps maintain cerebral venous pressure within normal values. IJV collapse during deep inspiration might be due to lower IJV intramural venous pressure. Because the venous wall is compressible and sensitive to changes in the transmural pressure gradient, the IJV lumen can vary with different phases of respiration [29]. During inspiration, the negative intrathoracic pressure increases IJV flow velocity and, according to Bernoulli’s principle, this leads to a reduction in the IJV lumen [20,30]. Reduced venous blood volume in IJV has been shown to reduce venous pressure at the baseline, making IJV more susceptible to inspiration-induced Bernoulli’s effect and causing significantly greater lumen decrement during deep inspiration [20]. Furthermore, age-related changes in the IJV reduce its ability to compensate for increased transmural pressure, leading to a reduction in distensibility and thereby predisposing the cerebral venous system to venous hypertension [31]. This study used abnormal IJV collapse in response to deep inspiration as a measure of distensibility. Although IJV collapse may have a postural dependency due to Bernoulli’s principle, all participants were asked to stay in a supine resting state and not to strain during breath-holding to avoid increasing intrathoracic pressure. In addition, we defined IJV collapse as the CSA of IJV shrinking to less than 1/3 of its original size during deep inspiration rather than the collapsibility bias of low venous flow.

MMSE is commonly used as a screening tool for cognitive impairment, with a score of 24–30 generally indicating no cognitive impairment [32,33]. Although this threshold can vary depending on level of education, we adopted the widely used cut-off of 24 points to avoid complex definitions that might obscure the focus of this study. For further clarification, we compared neuropsychological test results between participants with impaired IJV drainage and those in the control group. Our findings show that the impaired IJV drainage group scored lower than the control group across all neuropsychological tests used in this study. Although the MMSE scores of both groups were still within normal values, there were abnormal differences in MoCA scores. Among the neuropsychological tests, the MMSE had a sensitivity of 78% for AD, while the MoCA detects nearly 90–100% of AD [22,34]. Since the subjects in this study had a higher average educational level than the general population, a simple test such as MMSE is expected to yield relatively higher scores, and this test may not effectively distinguish between similar groups. MoCA is more sensitive and better at detecting abnormalities which contribute to the differences in scores between the two groups [33]. To our knowledge, cognitive function scales cannot identify pre-AD patients who will progress to AD in the future. Therefore, the proposed method for detecting impaired IJV drainage may effectively supplement MMSE or MoCA for enhanced cognitive screening. In our study, these neuropsychological tests revealed only subtle changes that do not necessarily indicate pathological decline. These differences should be further investigated through longitudinal studies to determine whether they translate into clinically meaningful cognitive impairment.

Cerebral venous congestion is a clinical state secondary to relative reduction in venous outflow in the brain. Hemodynamic alterations of the cerebral perfusion have been observed in patients with extracranial venous drainage abnormalities [35,36], whereas the patients with transverse sinus hypoplasia exhibited significantly lower outflow in bilateral IJV [6]. Severe IVJ insufficiency may lead to hypoperfusion of the brain parenchyma, resulting in brain cell degeneration [35]. Although the impact of IJV insufficiency on brain hemodynamics is not yet well understood, altered vascular structure and compliance induced by cerebral venous congestion are assumed to play crucial roles. Increased cerebral venous pressure can decrease cerebral blood flow, promoting local ischemia in the white matter and exacerbating pathological remodeling of the cerebral venules, leading to neurodegeneration and cognitive decline [3,10,37,38]. Some studies have suggested alterations in cerebral venous circulation may play a significant role in the development of brain pathologies in older individuals, including microhemorrhages, blood–brain barrier disruption, and perivascular inflammation [3,39,40]. Ultrasound-based studies have demonstrated age-related hemodynamic dysfunction with increased IJV diameter, reduced flow and increased reflux [4,27]. In summary, IJV insufficiency may be associated with neurodegeneration and cognitive decline in the elderly. While aging itself influences both cognitive function and IJV variability, our study found that, after adjusting for potential confounders through multivariate analysis, older age remained statistically significant but had relatively minor effects. In contrast, impaired IJV drainage was strongly and significantly associated with cognitive impairment.

Studies have shown that the prevalence of JVR increases with age, and the severity of JVR-associated white matter lesions is age-dependent, likely due to age-related degenerative changes in the venous valves [8]. JVR may compromise cerebral venous return, with retrogradely transmitted venous pressure reaching the cerebral venous system and influencing cerebral blood flow [8]. JVR not only causes retrograde flow but may also impair IJV outflow, potentially exacerbating brain dysfunction through increased retrograde venous pressure. VM is commonly used to evaluate JVR, which has been implicated in cognitive impairment. Although no abnormal venous reflux in the J2 or J3 segments was observed among the participants in this study, we did not implement VM to identify JVR cases and further confirm impaired IJV drainage because participants with cognitive impairment were considered unable to comply with the required VM procedure. Without MV-induced JVR, lower TAMV and lumen collapse alone might not fully explain IJV insufficiency. However, our results still show an association between impaired IJV drainage and cognitive impairment. Further studies are needed to validate these findings.

The present study has several limitations. First, the sample size is small. Secondly, there are no ideal fixed locations for measuring the CSA and TAMV in the J2 and J3 segments. To minimize bias, we measured the CSA and TAMV at the widest available lumen of the J2 and J3 segments. Our venous ultrasound examination followed standard protocols and was conducted by experienced technicians with extensive publications on the use of jugular venous ultrasound for diagnosis of various diseases [4,6,20]. Currently, there is no gold standard for the diagnosis of IJV pathologies, and assessment relies on a combination of various imaging modalities and criteria [11,41]. Since there are no standard reference ranges for impaired IJV drainage and collapse of IJV lumen in the literature, we sought to include the ultrasound data of study participants. Third, although we prospectively collected data from neuropsychological tests and ultrasound studies, the data were reviewed retrospectively, which may have introduced bias. While sleep apnea syndrome, cardiac function, or lifestyle factors such as smoking, alcohol consumption and dietary patterns may impact IJV drainage and cognitive function, this data was difficult to collect during routine evaluations in memory clinics and thus was not included in analysis. Similarly, while this study included routine CT and MRI evaluations, it lacked access to additional imaging to assess brain perfusion changes.

## 5. Conclusions

Impaired outflow in the bilateral IJV was found to be associated with impaired cognition. The impaired IJV drainage group also scored lower than the control group across all neuropsychological tests used in this study. Overall, these findings indicate that impaired IJV drainage is associated with global cognitive decline, though it remains unclear whether this association is merely correlative or indicative of a causal relationship between these pathologies. Longitudinal comparative analyses and/or comparisons across different subtypes of MCI could provide more definitive evidence. A better understanding of the underlying etiology may help identify early biomarkers of cognitive decline, potentially enabling therapeutic and preventive interventions before significant neurological damage occurs.

## Figures and Tables

**Figure 1 diagnostics-15-01427-f001:**
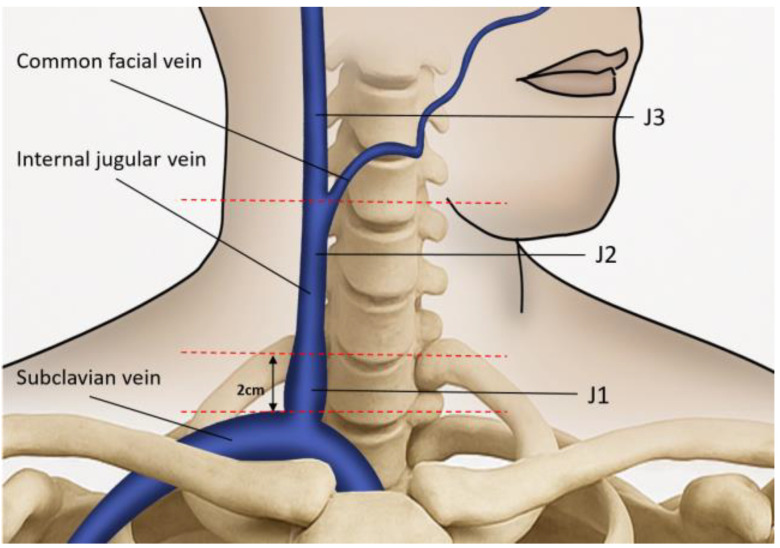
The ultrasound probe is placed over the internal jugular vein (IJV) to measure the time-averaged mean velocity and the lumen cross-sectional area. The anatomical landmark for J1 is located 2 cm above the upper edge of the clavicle. The junction between J2 and J3 is located where the common facial vein drains into the IJV. The segment of the IJV above this junction is referred to as J3, while the segment below it is referred to as J2.

**Figure 2 diagnostics-15-01427-f002:**
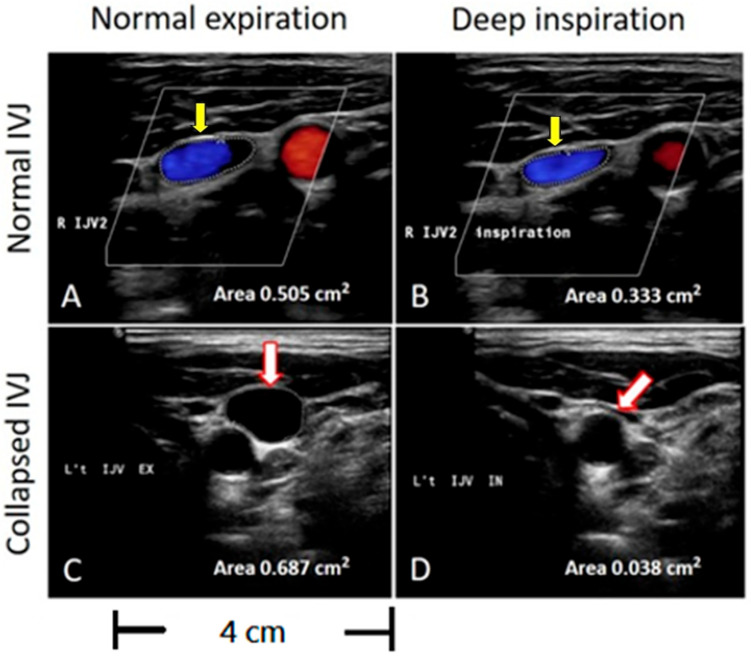
Evaluation of the CSA of the IJVs during inspiration on sonographic B-mode imaging. The circled area represents the cross-sectional area of the IJV. (**A**,**B**) Yellow arrows indicate no significant lumen collapse during deep inspiration compared to normal expiration, with an area change from 0.505 cm^2^ to 0.353 cm^2^. (**C**,**D**) White arrows indicate that the IJV shrinks to less than one-third of its original size during deep inspiration, with an area change from 0.687 cm^2^ to 0.038 cm^2^.

**Figure 3 diagnostics-15-01427-f003:**
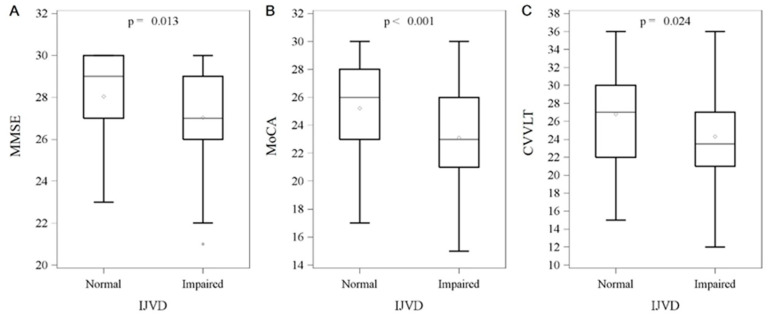
Between-group comparisons of impaired internal jugular vein drainage and intelligent test. Data are presented as the median with IQR, and 50% of study subjects are represented between the upper and lower bars. (**A**) MMSE (27 (IQR, 26~29) vs. 29 (IQR, 27~30), *p* = 0.013), (**B**) MoCA (23 (IQR, 21~26) vs. 26 (IQR, 23~28), *p* < 0.001), (**C**) CVVLT (23.5 (IQR, 21~27) vs. 27 (IQR, 22~30), *p* = 0.024). Abbreviation: IJVD: internal jugular vein drainage; IQR: interquartile range; MMSE: Mini-Mental State Examination; MoCA: Montreal Cognitive Assessment; CVVLT: Chinese Version of the Verbal Learning Test.

**Table 1 diagnostics-15-01427-t001:** Demographic data and neuropsychological test scores in 106 participants.

	All	Cognitive Impairment	Normal Cognition	*p*-Value
	*n* = 106	*n* = 50	*n* = 56	
Age (years old)	71.5 ± 8.0	73.4 ± 7.0	69.8 ± 8.5	0.0503
Gender (F/M)	67/39	33/17	34/22	0.3732
Education (years)	13.1 ± 3.8	11.3 ± 4.5	14.4 ± 2.9	0.0003 *
BMI (kg/m^2^)	24.3 ± 3.2	23.9 ± 3.1	24.7 ± 3.2	0.2195
Hypertension (%)	51 (48%)	22 (44%)	29 (52%)	0.4232
DM (%)	20 (19%)	12 (24%)	8 (14%)	0.2019
Dyslipidemia (%)	42 (40%)	14 (29%)	28 (50%)	0.0208 *
CKD (%)	4 (4%)	1 (2%)	3 (5%)	0.3652
Thyroid dysfunction(%)	15 (14%)	7 (14%)	8 (14%)	0.9664
History of stroke (%)	11(10%)	4 (8%)	7 (13%)	0.4482
History of MI (%)	1 (1%)	0	1 (2%)	0.3424
MoCA	24.3 ± 3.4	21.1 ± 2.4	26.9 ± 1.4	<0.0001 *
MMSE	27.6 ± 2.1	26.2 ± 1.9	28.9 ± 1.2	<0.0001 *

Abbreviation: BMI: body mass index; DM: diabetes mellitus; CKD: chronic kidney disease; MI: myocardial infarction; MoCA: Montreal Cognitive Assessment; MMSE: Mini-mental state examination. * *p* < 0.05.

**Table 2 diagnostics-15-01427-t002:** Comparisons of the TAMVs and CSA collapse of the IJVs between the patients with cognitive impairment and control subjects.

TAMV, cm/s	Cognitive Impairment (*n* = 50) No. (%)	Normal Cognition (*n* = 56) No. (%)	*p*-Value
LJ2 < 4 cm/s	17 (34%)	14 (25%)	0.3092
LJ3 < 4 cm/s	17 (34%)	10 (18%)	0.0569
RJ2 < 4 cm/s	5 (10%)	5 (9%)	0.8342
RJ3 < 4 cm/s	6 (12%)	1 (2%)	0.0321
LVV < 0.05 cm/s	11 (22%)	14 (26%)	0.6397
RVV < 0.05 cm/s	7 (15%)	8 (15%)	0.9425
LJ2 and LJ3 < 4 cm/s	15 (30%)	8 (14%)	0.0501
RJ2 and RJ3 < 4 cm/s	4 (8%)	1 (2%)	0.1319
**Impaired outflow** *	17 (34%)	9 (16%)	0.0322
CSA collapse **			
LJ2	18 (36%)	13 (23%)	0.1486
RJ2	10 (20%)	6 (11%)	0.1825
**Impaired IJVD** ***	34 (68%)	17 (30%)	0.0001

Abbreviation: L: left; R: right; J: jugular vein; VV: vertebral vein; IJVD: internal jugular vein drainage; * impaired outflow: TAMVs < 4 cm/s at either side of the internal jugular vein; ** collapse: cross-sectional lumen (CSA) in IJV shrinks to less than 1/3 of its original size after inspiration; *** impaired IJVD: TAMVs < 4 cm/s at either side of the internal jugular vein (LJ2/LJ3/RJ2/RJ3), or collapse at J2 segment on either side.

**Table 3 diagnostics-15-01427-t003:** Logistic regression model of the confounding variables associated with cognitive impairment.

	Univariate Analysis			Multivariate Analysis ^#^		
	OR	95% CI	*p*-Value	OR	95% CI	*p*-Value
Age	2.46	1.02~5.92	0.0445	1.07	1.00~1.13	0.0394
Education	0.79	0.70~0.90	0.0002	0.76	0.67~0.88	0.0001
MCA PI	12.35	1.12~136.5	0.0403			
Dyslipidemia	0.39	0.17~0.87	0.022			
Impaired IJVD	2.78	1.00~7.71	0.0495	3.84	1.33~11.09	0.0130

Abbreviation: MCA: middle cerebral artery; PI: pulsatility index; IJVD: internal jugular vein drainage; OR: odds ratio; CI: confidential interval. ^#^ Backward stepwise selection.

## Data Availability

Data are unavailable due to privacy.

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
