# Peer review of "Ultrasound Evaluation of Internal Jugular Venous Insufficiency and Its Association with Cognitive Decline"

_diagnostics, 2025, doi:10.3390/diagnostics15111427_

Round 1
Reviewer 1 Report
Comments and Suggestions for Authors
Dear Authors,
This is a well-written article about ultrasound (USG) evaluation of internal jugular vein (IJV) drainage. While interesting, several points need addressing:
1) Unless it is an RCT, a study cannot generally "clarify" an association, hence please re-phrase the aim of the study
2) The methodology needs more explaining: Is this a cross-sectional, cohort, or ambivalent study? How was sampling done? Vascular dementia subjects were excluded via what method? Was it with an MRI/CT scan? How many years did the ultrasound technician have under his/her belt?
I know that at the end of the discussion section, the authors address these issues, but they should be made more transparent and comprehensive in the methodology section.
3) Some figures could be added on how the ultrasound probe was placed and oriented to explain section 2.3. better
4) Please provide a cut-off score for MoCA and CVVLT, the same as how the authors provided the score for MMSE. Furthermore, MMSE scores are divided based on the level of education (i.e., those with lower education levels have a lower MMSE cut-off score), do clarify.
5) The authors mention in the methodology section that this is a "stepwise regression analysis (line 167)" yet in the result section, this is a "backward stepwise selection (line 221)". Please be consistent and precise with the methodologies used.
6) Please provide a CONSORT diagram of the patient recruitment
7) Please do address the inherent statistical downfall of creating a cut-off based on your own data (Lines 219-221).
Comments on the Quality of English LanguageSome sentences are awkward, for example:
Line 181: "Furthermore, the Spearman correlation coefficient between the basic information of the research sample and its neuropsychological test scores..."
Line 357: "Ultrasound laboratory"
I strongly suggest sending this manuscript to a professional English proofreader, as grammatically, there are no issues. This problem is more of a Taiwanese-English phrasing where the Mother Tongue influences the English language. Hence, an author who is proficient in the English language will not help here, as the grammar is perfectly fine (and not the main issue here).
Reviewer 2 Report
Comments and Suggestions for Authors
Dear authors,
It was a pleasure to read your well written manuscript, discussing the association between cognitive function impairment and IJV outflow/lumen collapse.
The employed methodology seems appropriate, and the statistical analysis is sound.
The provided graphical presentations are also adequate and easily understandable.
However I strongly believe that some improvements may be applied.
- Line 67 please correct the normal hydrocephalus to normal pressure hydrocephalus
- Please try not to repeat information through the text extending without any significant reason the number of words (eg line 175 - between may 2022 and may 2023 has already been stated.)
- Please mention if you have performed a sample size estimation prior to conducting the study.
-The conclusion should be more concise and less descriptive.
-Lines 269-270 I could not understand the "(data not shown)" the sentence is confusing. Please rephrase. (Probably authors meant that not supported by their data?)
Comments on the Quality of English Language
Some parts need to be copyedited as they are confusing or difficult to be understood.
Round 2
Reviewer 1 Report
Comments and Suggestions for Authors
The authors have successfully addressed all the issues
Reviewer 2 Report
Comments and Suggestions for Authors
Dear authors,
I would like to thank you for implementing my suggestions.
Best regards,